# Effects of interoceptive accuracy in autonomic responses to external stimuli based on cardiac rhythm

Yuto Tanaka[1]*, Yuri Terasawa[2], Satoshi Umeda[2]

1 Global Research Institute, Keio University, Minato-ku, Tokyo, Japan, 2 Department of Psychology, Keio University, Minato-ku, Tokyo, Japan

☯ These authors contributed equally to this work.
* yuto.tanaka@keio.jp

**Data Availability Statement:** All relevant data are within the paper and its Supporting Information files.

**Funding:** Japan Society for the Promotion of Science (JSPS) Award Number: JP16H03740 |

## Abstract

Interoceptive accuracy is an index of the ability to perceive an individual's internal bodily state, including heartbeat and respiration. Individual differences in interoceptive accuracy influence emotional recognition through autonomic nervous activity. However, the precise mechanism by which interoceptive accuracy affects autonomic reactivity remains unclear. Here, we investigated how cardiac reactivity induced by a non-affective external rhythm differed among individuals, using a heartbeat counting task. Because individuals with poor interoceptive accuracy cannot distinguish an external rhythm from their cardiac cycles, it has been hypothesized that the interoceptive effect on heart rate works differently in individuals with good interoceptive accuracy and those with poor interoceptive accuracy. Study participants observed a visual or auditory stimulus presented at a rhythm similar to the participants' resting heart rates. The stimulus rhythm was gradually changed from that of their resting heart rate, and we recorded electrocardiographs while participants were exposed to the stimuli. Individuals with good interoceptive accuracy exhibited a deceleration in heart rate when the rhythm of the auditory stimulus changed. In contrast, in the group with poor interoceptive accuracy, the heart rate decreased only when the stimulus became faster. They were unable to distinguish the rhythm of their own heartbeat from that of the external rhythm; therefore, we propose that such individuals recognize the stimuli at the pace of their heart rate. Individuals with good interoceptive accuracy were able to distinguish their heart rates from the external rhythm. A modality difference was not observed in this study, which suggests that both visual and auditory stimuli help mimic heart rate. These results may provide physiological evidence that autonomic reactivity influences the perception of the internal bodily state, and that interoception and the autonomic state interact to some degree.

## Introduction

Over the past two decades, it has become widely known that we perceive not only our outside environment but also the information inside our bodies. The ability to perceive internal body

Recipient: Satoshi Umeda Japan Society for the Promotion of Science (JSPS) Award Number: JP19K21819 | Recipient: Satoshi Umeda Scientific Research on Innovative Areas (Ministry of Education, Culture, Sports, Science and Technology) Award Number: JP18H05525 | Recipient: Satoshi Umeda.

**Competing interests:** No authors have competing interests.

states, such as heartbeat and respiration, is known as interoception [1, 2]. This term was first proposed by Sherrington [3] to distinguish interoception from exteroception (perception of the external environment) and proprioception (perception of muscular movement). Interoception also involves how people perceive their bodies and gain self-awareness [4–6]. Interoception plays an essential role in maintaining homeostasis, which helps maintain physical and mental health; failure to maintain a suitable homeostatic state can result in psychiatric disorders [7, 8]. Individual differences in how accurately a person perceives his/her own body state is called interoceptive accuracy. Individuals with good interoceptive accuracy can track internal bodily sensations as accurately as objective measures. In contrast, individuals with poor interoceptive accuracy have no or limited perceptions of their internal bodily sensations from objective measures [9].

Individuals with good interoceptive accuracy experience higher arousal from emotion-inducing pictures than individuals with poor interoceptive accuracy [10–12]. Interoceptive accuracy is also associated with other emotional aspects, such as detecting the emotions of others [13], decision-making involved in gambling [14], social decision-making [15], and attentional task performance [16]. These studies hypothesize that not only the state of the body but also the perception of the internal body is important when assessing psychological phenomena; however, the researchers did not focus on the autonomic response itself. Autonomic reactivity was assessed when participants were exposed to an external stimulus, based on changes in the cardiac r–r interval and the distance between the two r waves on the electrocardiogram [17, 18]. Because interoception is associated with maintaining homeostasis, which is required to monitor the autonomic nervous system [8], investigating the association between autonomic reactivity and interoceptive accuracy is necessary to understand how interoception is associated with the mental state and physiological state of the body.

Previous studies have proposed that interoception is associated with autonomic activity itself, although other studies do not acknowledge this hypothesis [14, 18]. To further elucidate this issue, an investigation of autonomic responses to non-affective, neutral stimuli is required. Researchers may focus on emotional stimuli because they elicit distinctive effects on both the mental and physiological states. Although this association is important, various factors such as emotional valence and arousal, subjective recognition of emotional experience, and personality factors are intertwined. Additionally, because interoceptive accuracy also involves emotional processing [19], it is difficult to directly observe the association between autonomic reactions and interoception. By using non-affective, neutral stimuli, it may be possible to observe the association between cardiac response and individual differences in interoceptive accuracy. Several studies have indicated that some non-affective stimuli can cause slight changes in heart rate. For example, heart rate was shown to decrease when participants listened to an auditory stimulus with a decreased cardiac tempo, which we call decreased cardiac tempo [20]; thus, heart rate can be affected by the rhythm of a stimulus. Another study showed that heart rates decreased when participants listened to music with a faster rhythm than their resting heart rate, called increased cardiac tempo [21]. In a different study, although individual differences were noted, participants' heartbeats were synchronized to a musical rhythm when the rhythm gradually decreased [22].

External stimuli in which the rhythm approximates the heartbeat are often treated as a representation of the heartbeat in medical settings or research on interoception; a typical example is the heartbeat discrimination task. Similar to the heartbeat counting task, this task is treated as a measure of interoceptive accuracy [23]. Although it is has been pointed out that this task is difficult, it enables observation of individual differences, and both heartbeat counting task and heartbeat discrimination task are proper indicators of interoceptive accuracy [9, 24]. Therefore, the ability to distinguish an external stimulus's tempo from that of the

heartbeat is essential for accurately capturing interoception. A previous study has shown that the perception of cardiac interoception is sensitive to external rhythms [25]. The ability to discriminate between one's heartbeats and external stimuli by interoceptive sensations depends on ignoring other exteroceptive information, thereby allowing appropriate mental state adjustments. In this case, the autonomic response may vary depending on interoceptive accuracy.

Recent studies have suggested that the rhythm of external stimuli affects the autonomic nervous system when the rhythm is similar to an individual's heart rate; infants feel relaxed when they listen to the sounds of their mothers' heartbeats [26]. Moreover, adults feel most relaxed when they listen to a tempo similar to the cardiac rhythm, which is approximately 70 to 80 beats per minute [27]. These results suggest that rhythms similar to an individual's heart rate affect both the mental state and autonomic nervous system. Another study showed that participants were better at identifying their heartbeat than another individual's heartbeat [28].

In the present study, the external stimulus rhythm was initially based on the participants' resting heart rate and then altered to become either faster or slower. We measured how the heart rate changed when the external rhythm deviated from the participant's heart rate. Previous studies using periodic rhythm instead of music have only used stimuli with a decreasing tempo; we extended the tempo range to study the effect of stimuli with increasing tempos. Previous studies using periodic rhythm instead of music have only used stimuli with a decreasing tempo [20, 22]; we extended the tempo range to study the effect of stimuli with increasing tempos. A previous study has claimed that a decrease in the heart rate is observed when participants are exposed to a stimulus with a decreasing tempo [29]; therefore, a different effect may be seen when the tempo increases. The effect may be akin to listening to a tempo during exercise, where the psychological effects are different from those in a resting state [30].

An increased tempo might have a different effect on decreasing cardiac tempo. Studies have not been consistent in elucidating how the heart rate changes when stimuli become faster [31, 32]. Participants often perceive the heartbeat to be faster, but rarely perceive it to be slower. Therefore, interoceptive accuracy may be influenced only by a faster stimulus. For individuals with poor interoceptive accuracy, an accelerating stimulus is expected to increase the heart rate. Because individuals with good interoceptive accuracy will notice this increase, their heart rate will not change.

The modality of exteroception may affect autonomic reactivity. Several experiments on interoception have used feedback from participants' heartbeats as auditory stimuli [2, 25, 26]. Other studies have used visual stimuli as feedback for autonomic activity, specifically in the study of biofeedback [33]. In addition, the simultaneous presentation of visual and auditory stimuli may produce changes that are different from those of a single rhythm. For instance, reaction times to the stimulus were accelerated when multiple modalities were presented simultaneously; this is termed intersensory facilitation [34]. Many other studies have shown that multisensory integration affects mental processing [35]; however, the effect of multisensory integration on the tempo of the heartbeat has not been well discussed. Therefore, we aimed to observe whether the visual or auditory senses respond preferentially. For example, in the sound-induced flash illusion, visual information is influenced by auditory information [36]. Determining the differences in response to modality will reduce the extraneous variables of prior experiments and elucidate how individuals detect exteroceptive rhythms that can influence their biological rhythm.

The present study also aimed to determine the association between individual differences in interoceptive accuracy and autonomic reactivity induced by periodic non-affective stimuli. Determining this association is considered beneficial for a comprehensive understanding of the mechanisms underlying the effect of external state on our bodies. We used visual and auditory stimuli to compare the differences in modality. Because it is easier to perceive interval

changes in auditory stimuli [37], we hypothesized that auditory stimuli would show more autonomic reactivity than visual stimuli. The rhythm of the stimulus was accelerated and decelerated to clarify whether autonomic changes reflected the direction of exteroceptive stimuli. We hypothesized that differences in interoceptive accuracy affect changes in participants' heart rates while observing periodic stimuli, and that the degree of change depends on the modality of the stimulus.

## Materials and methods

### Participants

This study was approved by the Keio University Research Ethics Committee (No. 15012–1). We performed a priori power analysis for the sample size estimation for *F*-test based on our study design, two groups with three measurements, using GPower 3.1 [38]. Assuming that this study has a medium effect size, we used 0.25 based on Cohen's criteria [39]. With an alpha = .05, power = 0.80, and correlation = 0.50, it indicated that a sample size of at least 28 was necessary. A total of 43 graduate and undergraduate students at Keio University (20 men and 23 women) participated in this experiment. The mean age of participants was 21.70 years (*standard deviation* = 1.26). Four participants were excluded from the analysis because of problems with the apparatus during the experiments. Before the experiment, all participants provided informed consent according to the ethics code of the Department of Psychology at Keio University.

### Apparatus

Electrocardiographs (ECGs) were recorded while the participants performed three tasks (MP-150; Biopac Systems, Santa Barbara, CA, USA). The silver/silver chloride electrodes were attached to the back of the participants' right hands and ankles of both legs. The participants' resting heart rates were measured before performing the tasks. Physiological data were analyzed using AcqKnowledge (Biopac Systems, Santa Barbara, CA, USA). The stimuli were presented using Presentation (Neurobehavioral System, CA, USA).

### Tasks

In this study, to investigate the influence of interoceptive accuracy on heart rate, participants performed three tasks. We conducted a heartbeat counting task, which has previously been used to measure individual differences in interoceptive accuracy [40]. Because it has been suggested that heartbeat counting may result in a good performance by simply estimating the time [24], we also conducted a time perception task that had a similar procedure to the heartbeat counting task. This step ensured that the score of the heartbeat counting task was not the same as the accuracy of the time estimation. For the main task, non-affective visual and auditory stimuli were presented to make slight changes to the participants' heart rates.

For the heartbeat counting task, participants were instructed to silently count their heartbeat without touching their body or any other place that might facilitate the perception of heartbeats. The trials were 2 × 25 s, 2 × 35 s, and 2 × 45 s in length. All participants performed six trials. The beginning and end of each trial were displayed on a computer monitor. At the end of each trial, the participants were required to report their counts on a keyboard.

Interoceptive accuracy was calculated based on the difference between the actual heart rate and the number of heartbeats reported by the participant. We calculated the score of the heartbeat counting task using the following formula: (1/6 Σ (1 − (|actual heartbeats − reported

heartbeats|/actual heartbeats)). Scores closer to 1 indicated that the participants had accurate interoception.

We also conducted a time estimation task. Each participant was instructed to count the time in seconds. The trials were 2 × 25 s, 2 × 35 s, and 2 × 45 s long, the same length as in the heartbeat counting task. All participants performed six trials. The beginning and end of each trial were displayed on a computer monitor. Participants reported their answers using a keyboard. The time estimation accuracy was calculated based on the formula for the heartbeat counting task as follows: 1/6 (1 –[Σ |actual time—reported time|/actual time]). A score closer to 1 indicates that the participant had an accurate time estimation.

During the main experimental task, a pure tone, a red circle, or both stimuli were presented for a specific time interval. A pure tone of 1000 Hz was presented using headphones, while a red (R255, G0, B0) circle was displayed at the center of a computer monitor. After the presentation of stimuli for 200 ms, a blank screen was presented for the remainder duration, as if they were flashing at the interval of the participant's heart rate. For this task, the participants were instructed to watch or listen to the presented stimuli. Participants were not instructed to respond to these stimuli.

For the first 30 s, visual, auditory, or both stimuli were presented for an interval that corresponded to the heartbeat of the participant measured during the resting period. If the participant's baseline IBI was 750 ms, the stimulus was presented for 200 ms, followed by a blank screen that had a 550-ms duration. For the next 30 s, the interval duration was modified so that the duration of the blank display was either 30 ms shorter or 30 ms longer. This modification was applied three times across each trial; hence, the final trial was either 90 ms shorter or 90 ms longer than the first interval (Fig 1). Participants conducted six types of conditions:

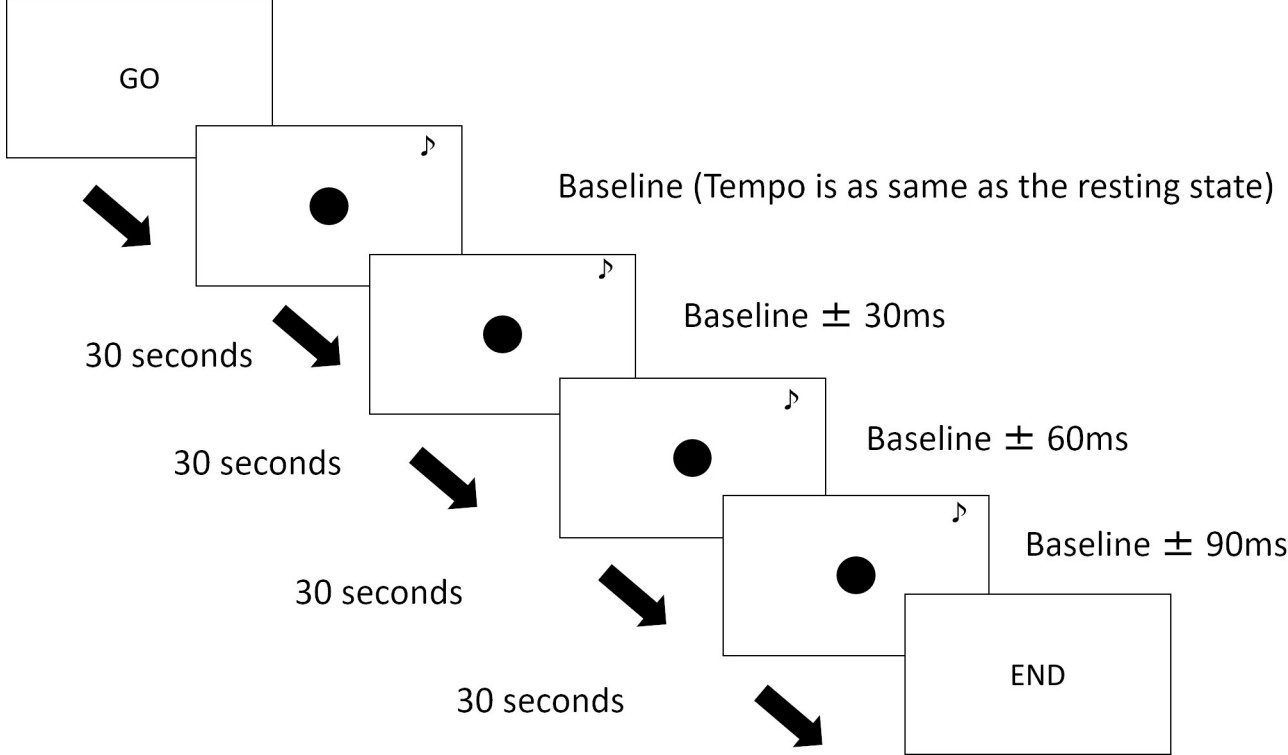

**Fig 1. Sequence of the experimental task.** In each trial, the stimuli were presented for 200 ms, followed by a blank screen for the rest of the interval. After 30 s, the stimulus became either 30 ms faster or 30 ms slower. This change in speed happened three times during the task; hence, the speed had changed by 90 ms at the end of the trial. Each trial lasted approximately 2 min.

increasing frequency of visual stimuli, decreasing frequency of visual stimuli, increasing frequency of auditory stimuli, decreasing frequency of visual stimuli, increasing frequency of visual and auditory stimuli, and decreasing frequency of visual and auditory stimuli. These conditions were combined into one group, and each of the six conditions was performed twice, as shown in Table 2. The participants were not informed that the stimulus rhythms were modified during the trial. The ECGs were recorded while the participants completed the trials.

We calculated the average IBI for the baseline period, the rhythm that increased from baseline, and the rhythm that decreased from baseline. Because we wanted to determine the difference in IBI when the speed of the stimuli was increased or decreased, we averaged the three types of speeds (30 ms, 60 ms, and 90 ms) and created a single variable.

### Procedure

We combined these three tasks into one experiment. In one session, each participant completed the heartbeat counting task, the main task, the time perception task, and the main task (Fig 2). The task conditions (the duration of the heartbeat perception and time estimation tasks and the stimulus type of the main task) were chosen randomly using a computer. Each participant completed all six conditions, and all participants completed both sessions with heartbeat counting and time estimation tasks. Participants took a break between the sessions for 10 minutes. ECG data were collected during all tasks. The entire experiment took 1 h, including the time required to set up the apparatus and break times.

## Results

### Heartbeat counting task

Among the 39 participants, the average interoceptive accuracy was 0.73 ($SD$ = 0.16). Individual data are shown in S1 File. We used a median split method, as used in a previous study [41], to divide participants into a good interoceptive accuracy group and a poor interoceptive accuracy group. Based on the score in this task, the participants were categorized into two groups based on good interoceptive accuracy ($n$ = 19) or poor interoceptive accuracy ($n$ = 19); we excluded participants with an average score on the heartbeat counting task. The average interoceptive accuracy of participants in the good interoceptive accuracy group was 0.86 ($SD$ = 0.12), and the average interoceptive accuracy of participants in the poor interoceptive accuracy group was 0.60 ($SD$ = 0.07). There was a significant difference in interoceptive accuracy between the

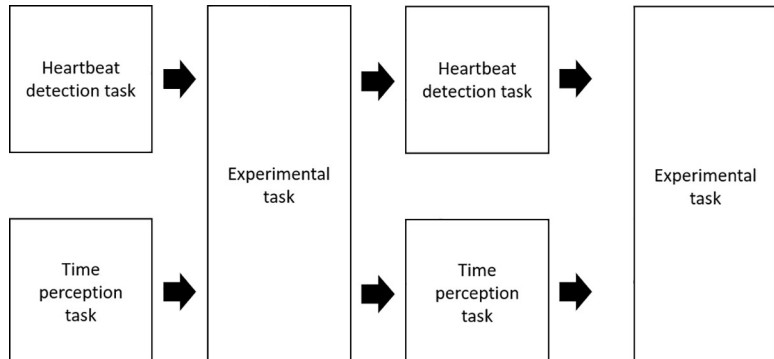

**Fig 2. Sequence of the experiment.** In one session, participants completed either the heartbeat perception task alternating with the experimental task or the time estimation task alternating with the experimental task. We randomized the interval in the heartbeat perception and time estimation tasks and the type of stimulus and direction of speed change in the experimental task. The session order was counterbalanced across participants.

**Table 1. Scores in the heartbeat counting and time estimation tasks in good heartbeat perceivers and poor heartbeat perceivers (standard deviation).** There was a significant difference in the scores in the heartbeat counting task, but not in the time estimation task, between the two groups.

|  | Good perceiver | | Poor perceiver | | t | p | |
|---|---|---|---|---|---|---|---|
| Heartbeat counting task | .86 | (0.12) | .60 | (.07) | 10.15 | < .01 | ** |
| Time perception task | .82 | (0.14) | .77 | (.10) | 1.49 | .15 | |

**p < .01.

two groups ($t$ (36) = 10.15, $p < .01$, $d = 2.65$, Table 1). The average heart rates in the good interoceptive group and the poor interoceptive accuracy group were 78.05 and 79.78%, respectively. There were no significant differences in the baseline heart rate between the two groups ($t$ (36) = 0.76, $p = .31$).

## Time estimation task

The average time estimation accuracy among all the participants was 0.80 ($SD = 0.12$). There was no difference in the time estimation accuracy between the good interoceptive accuracy group and the poor interoceptive accuracy group ($t$ (36) = 1.49, $p = 1.44$). The score on the heartbeat counting task was not significantly associated with the score on the time estimation task ($r$ (39) = .26, $p = .11$).

## Experimental task

The average values of the IBI during the experimental tasks for each of the three conditions are listed in Table 2. We conducted a 3-way analysis of variance (ANOVA) to determine whether the difference in IBI was due to the difference in interoceptive accuracy (good or poor interoceptive accuracy), modality of the stimulus (auditory, visual, or compounding of auditory and visual), or the speed of the stimuli (baseline, faster than baseline, or slower than baseline). The degrees of freedom were adjusted using the Greenhouse–Geisser procedure.

We found an interaction between interoceptive accuracy and the speed of the stimulus ($F$ [1.92, 69.10] = 3.32, $p < .05$, pη2 = .10, CI—.01–.20). In the poor interoceptive accuracy group, heart rate increased when the stimulus speed was the increased cardiac tempo than when the stimulus speed was the decreased cardiac tempo, or when the stimulus speed was the baseline heart rate (Fig 3). In the good interoceptive accuracy group, the heart rate decreased when the stimulus speed was either faster or slower from the baseline (Fig 3). These results indicate that interoceptive accuracy is related to autonomic responses induced by non-affective rhythms.

## Discussion

This study aimed to determine whether interoceptive accuracy is associated with cardiac reactivity elicited by non-affective visual or auditory stimuli presented in a rhythm similar to the cardiac cycle. Individuals with poor interoceptive accuracy exhibited a change in heart rate

**Table 2. Interbeat intervals (standard deviation) while the participants observed rhythmic visual or auditory stimuli (sec).**

|  | Auditory stimuli | | Visual stimuli | | Visual and Auditory stimuli | |
|---|---|---|---|---|---|---|
|  | Good perceiver | Poor perceiver | Good perceiver | Poor perceiver | Good perceiver | Poor perceiver |
| Baseline Condition | 0.799 (0.079) | 0.772 (0.090) | 0.792 (0.076) | 0.774 (0.093) | 0.795 (0.076) | 0.779 (0.075) |
| Faster condition | 0.810 (0.072) | 0.790 (0.092) | 0.802 (0.077) | 0.781 (0.093) | 0.808 (0.076) | 0.788 (0.096) |
| Slower condition | 0.815 (0.078) | 0.775 (0.084) | 0.801 (0.071) | 0.780 (0.090) | 0.808 (0.075) | 0.776 (0.091) |

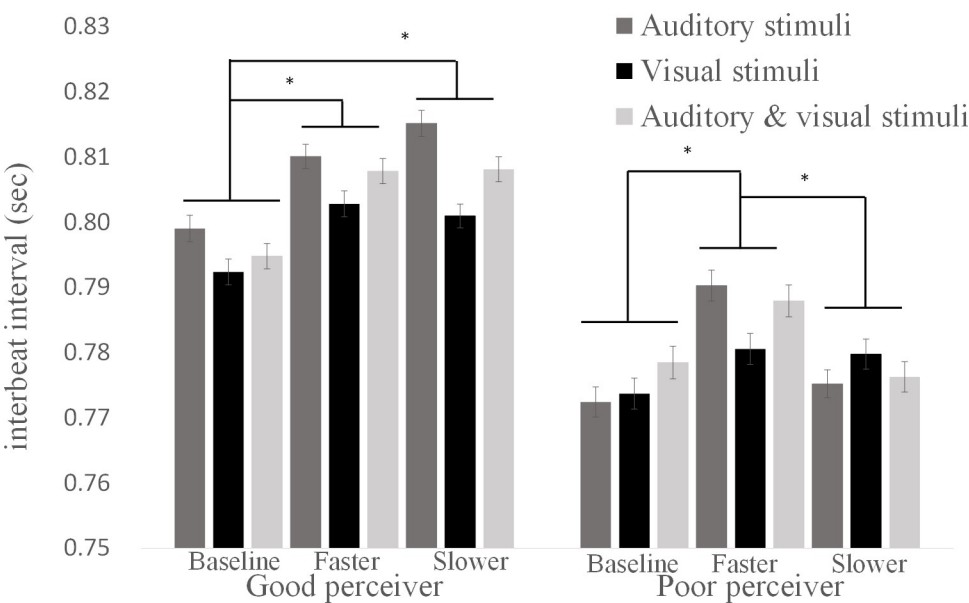

**Fig 3. Interbeat intervals (IBIs) while presenting external rhythmic stimuli.** In poor heartbeat perceivers, IBI increased when the speed of the stimuli was faster than their baseline heart rate. In good heartbeat perceivers, IBI increased when the speed of the stimuli was changed from their baseline heart rate. There were no significant differences among the modality of the stimulus.

when the rhythm of the stimulus was at an increased cardiac tempo. This tendency was not observed in individuals with good interoceptive accuracy. Instead, their heart rate became slower when the speed of the stimulus changed, regardless of whether it got faster or slower. The presumption is that these individuals implicitly recognize external stimuli as their heart rate when it is difficult for them to perceive their internal state.

The results imply that interoceptive accuracy affects autonomic reactivity and suggests that the efferent signal is affected by the afferent signal, resulting in an efferent signal that creates a new afferent signal; thus, the signals interact with each other [42, 43]. This process is important for maintaining a homeostatic state in which interoception plays a critical role [1]. Our results demonstrate that the heart rate is decreased by non-affective stimuli, a result that is consistent with the results of a previous study that used music or auditory stimuli with a rhythm similar to the participants' heart rates [20]. Notably, this effect was also observed when the external rhythm increased in the tempo. Previous studies have focused only on decreasing rhythms [21, 22]. An increased heart rate was observed in previous studies using music [44–46], but these studies were distinctly different from our study because they used a rhythm almost twice as fast as the heart rate. Our results suggest that a rhythm similar to the heart rate causes deceleration even when the stimulus is accelerated.

We suggest that part of the interaction between interoceptive accuracy and the autonomic response considered in this study is closely associated with an orienting response, that is, a response to a novel stimulus [47, 48]. In this study, exteroceptive stimuli began at a pace similar to the participant's heart rate and subsequently deviated from the heart rate, which could be perceived as a novel stimulus if this change in tempo was noticed. If this phenomenon was just a general orienting response to the novel states of stimuli, then the decreasing heart rate with longer r–r intervals would recognize the change in the speed of the external stimuli.

Perceiving changes in the auditory rhythm were considered a novel stimulus because participants perceived the changes in the rhythm more easily than the visual stimulus. Because

individuals with good interoceptive accuracy can discriminate the external stimulus from their heart rate, the auditory stimulus is more likely to be interpreted as a different signal from the internal signals. For participants with good interoception, stimuli may be distinguished as being different from the heartbeat yet remain heartbeat-like stimuli; this is consistent with the findings of previous studies that have been predicted to induce relaxation [22, 27]. Although it is difficult to evaluate the psychological effects of the stimuli in this study, we suggest that the parasympathetic nervous system became dominant during the viewing process. Conversely, this was only observed for individuals in the poor interoceptive accuracy group when the stimulus became faster. It may be easier to perceive one's heartbeat when it is noticeably faster, for example, after exercise [49]. Therefore, it is likely that the stimulus rhythm would be predicted to be the same as the heartbeat. Overall, the differences in r–r intervals were associated with the differences in interoceptive accuracy, and experimental manipulation confirmed the decrease in heart rate in individuals with poor interoceptive accuracy.

The results of the present study showed no difference in heart rate according to modality, contradicting the hypothesis. This suggests that both visual and auditory stimuli are useful for mimicking the heart rate and that cross-modal effects [50, 51] do not occur—even when they are presented simultaneously. Although the perceived ease of change was different depending on the type of stimulus, it was not affected by the type of stimulus, further suggesting that the ability to compare the rhythm and interoception is more important than the ease of conscious perception of the stimulus. Additionally, the findings of this study suggest that people implicitly use interoception to determine whether external stimuli are at the same tempo as the heart rate, corroborating the findings of an earlier study [52].

In this study, the speed of the external stimuli that participants were exposed to was determined by each participant's resting heart rate. Therefore, the external stimuli purposefully did not synchronize with the participant's actual heartbeat. When the heartbeat and auditory stimuli are synchronous, future predictions depend on when the auditory stimuli are presented [53], but synchronized stimuli cannot fix the IBI in the experiment. Because we measured the autonomic response caused by the exteroceptive stimuli, heart rate was influenced by the experimental manipulation of extending or shortening the IBI of the stimulus. Our results suggest that individual differences in interoceptive accuracy affect the autonomic responses. Therefore, presenting stimuli with a different rhythm from the participants' heart rate could help individuals with good interoceptive accuracy to distinguish the corroborating findings of an earlier study more easily.

Some limitations of our current work must be considered in the design of future studies. This study was only able to show a minimal effect size. Increasing the research size and increasing the sample size would be necessary to establish this effect as robust. The baseline in this study referred to the mean resting heart rate, which was not recorded at the beginning of the task. Therefore, the r–r interval could have been different from the baseline when the participants started their trials. Even if stimuli were presented synchronously with the heartbeat, it is unclear whether the participants would perceive this as a heartbeat, and fluctuations in the heartbeat could erase experimental manipulation changes. Because synchronizing with the heartbeat creates various constraints, this study aimed to make steady changes in the stimuli speed. Future studies may benefit from making the baseline rhythm closer to the heartbeat, for example, to better match the tempo of the stimuli to that of the heart rate at the start of the trial. In addition, the study did not compare the baseline heart rate and the condition in which the stimulus was presented. Although there was a significant difference between these states, the possibility that interoceptive accuracy is involved in switching the state with modality and the state without modality cannot be denied. In future studies, it will be important to increase the inter-trial interval and compare the results to baseline before stimulus presentation and

between trials. Moreover, there were only two trials in this study, and these were relatively small. Since only a single trial was conducted for each condition in the previous study [21, 45], it was considered more important to take as much time as possible for one trial than to increase the number of trials. In addition, a long experiment may result in increased fatigue of the participants, which may affect their heart rates. Therefore, in the future, increasing the number of trials by dividing the experiment into multiple days may produce results that are more consistent and demonstrate that interoceptive accuracy is related to autonomic reactivity. In addition, long-period autonomic responses were observed in the current experiments, but interoceptive accuracy might have a different effect during the rapid response induced by affective stimuli. In future studies, determining the autonomic response caused by a phase shift of exteroceptive stimuli from the actual heart rate will be important in assessing the association between interoception and autonomic reactivity.

It would also be helpful to determine whether autonomic activity influences conscious perception when changes in the IBI of the stimulus are recognized. Another limitation is that we only conducted the heartbeat counting task to measure interoceptive accuracy. Some studies suggest that the perception of heartbeat in the heartbeat counting task influences the belief that the person has in their heartbeat, and additional studies have reported that the heartbeat counting task does not correspond well to other measurements such as the heartbeat detection task [25, 53–56]. It may be important to measure interoception in several different ways, such as using a questionnaire that measures interoceptive sensibility [57]. While there is a robust relationship between emotional stimuli and internal receptive sensations, a solid scientific background is lacking since no emotional information is involved. The results of our study suggest that interoception commits to the perception of processing stimuli, as well as other exteroceptive sensations.

Although this study has some limitations, it is clear that interoceptive accuracy significantly affects the autonomic response induced by non-affective stimuli. This result suggests that our body interacts with external perception, which may help in determining the psychophysiological mechanisms of the body-mind interaction. Further studies are necessary to assess the explicit expression of these interactions in the mental state. Moreover, investigation of the association between interoception and physiological responses is required to understand how our body and mind interact with the environment.

## Supporting information

**S1 File. Interoceptive data and heart rate data of the participants.**
(XLSX)

## Acknowledgments

We would like to thank Editage (www.editage.jp) for English language editing.

## Author Contributions

**Conceptualization:** Yuto Tanaka, Yuri Terasawa, Satoshi Umeda.

**Data curation:** Yuto Tanaka, Yuri Terasawa.

**Formal analysis:** Yuto Tanaka, Yuri Terasawa, Satoshi Umeda.

**Funding acquisition:** Satoshi Umeda.

**Investigation:** Yuto Tanaka, Yuri Terasawa, Satoshi Umeda.

**Methodology:** Yuto Tanaka, Yuri Terasawa, Satoshi Umeda.

**Project administration:** Satoshi Umeda.

**Resources:** Yuto Tanaka.

**Software:** Yuto Tanaka, Satoshi Umeda.

**Validation:** Yuto Tanaka, Yuri Terasawa, Satoshi Umeda.

**Visualization:** Yuto Tanaka.

**Writing – original draft:** Yuto Tanaka.

**Writing – review & editing:** Yuto Tanaka, Yuri Terasawa, Satoshi Umeda.

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
