## [Decision Letter · Decision Letter 0]

8 Mar 2021

PONE-D-20-20821

Effects of Interoceptive Accuracy in Autonomic Responses to External Stimuli Based on Cardiac Rhythm

PLOS ONE

Dear Dr. Tanaka,

Thank you for submitting your manuscript to PLOS ONE. After careful consideration, we feel that it has merit but does not fully meet PLOS ONE’s publication criteria as it currently stands. Therefore, we invite you to submit a revised version of the manuscript that addresses the points raised during the review process.

Please read the comments of all the reviewers carefully and revise your manuscript accordingly.  Please do address the concerns of the reviewers such as major weakness in statistical analyses to report the findings with a systematical manner after the performance of rigorous statistical  analyses of the data and return significantly improved revision. 

We look forward to receiving your revised manuscript.

Kind regards,

Zhishun Wang, Ph.D.

Academic Editor

PLOS ONE

Journal Requirements:

Reviewers' comments:

Reviewer's Responses to Questions

**Comments to the Author**

1. Is the manuscript technically sound, and do the data support the conclusions?

Reviewer #1: Partly

Reviewer #2: No

2. Has the statistical analysis been performed appropriately and rigorously? 

Reviewer #1: No

Reviewer #2: Yes

3. Have the authors made all data underlying the findings in their manuscript fully available?

Reviewer #1: No

Reviewer #2: No

4. Is the manuscript presented in an intelligible fashion and written in standard English?

Reviewer #1: Yes

Reviewer #2: No

5. Review Comments to the Author

Reviewer #1: The study investigates if cardiac reactivity changes in response to manipulations in the external rhythm (auditory, visual and audio-visual changes). The authors divide two groups in poor and good perceivers according to the heartbeat counting task results.

The idea quite interesting, the main aim was to study such changes in response to non-affective external stimuli, however the study presents a few weaknesses both theoretically and methodologically:

1) The introduction and the hypotheses could do a better job. The style could be a little more colloquial, it seems more like bullets points.

It is unclear: i) the rational in having faster and slower conditions from baseline and why only the faster condition seems to be affected; ii) why three different modalities (one unimodal A and V and another bimodal AV) were tested and how a specific modality should be affected more or less than others. I wonder if using a single modality increasing the number of trials would have been a better way, rather than introduce two more modalities without very strong hypotheses and have few trials; iii) it seems that there was the aim to study differences in a group of people with high and low interoceptive accuracy that was not properly investigated at the step on statistical analysis (see point2).

I suggest to clarify all these points in the introduction/aims/hypothesis.

2) Statistical Analysis: It is unclear how the 3 modalities (A/V/AV) were analyzed in the first ANOVA. If the aim was to investigate the differences between participants with higher and lower interoceptive accuracy and the modalities, I think that a single ANOVA design with group (poor and good perceivers) as between factor; speed (baseline, faster, slower) and stimulus type (A, V, AV) as within levels; would have been a better design. I wonder whether there was any other difference in the results between the two groups that was not reported. It seems that the two groups differ in the slower conditions, at the least in two modalities (A and AV).

I would suggest to perform a between subjects ANOVA 2Group x 3Speed x 3Stymulus Type, as I explained above. Once this has been clarified it will help the decision on the validity of the results.

3) The method used is my biggest concern, it seems that the conditions (faster and slower) were manipulated according to the cardiac rhythm registered at baseline offline (prior the task execution). Therefore the external stimuli were not perfectly synchronized (slower or faster in 30 ms) with the real rhythm online at the time of the task execution. I fatigue to accept the validity of their results, as the real cardiac rhythm could have been changed during the task execution and not recorded, in this way the external stimuli could never adjust accordingly.

Add this as a limitation of the study.

Minor points:

1) Page 1 line 22: specify r-r

2) Page 5 line 110: I wonder if this is a typo and line 110 should be: (1/6 Σ(1 − (|actual heartbeats − reported heartbeats|/actual heartbeats)).

3) It is unclear the total number of trials: 6 trials for each condition (baseline - shorter and longer interval?) or 6 trials for each modality (A V AV) ? Please specify.

4) I would avoid any discussion relating brain activity as the study is purely behavioral. Please remove.

Reviewer #2: The authors investigate the cardiac reactivity in healthy participant by recording ECG during passive presentation of auditory, visual stimuli or both. Sensory stimuli followed a rhythm closely matching participants’ heartbeat and then gradually accelerating or decelerating. Participants were separately screened in order to evaluate their interoceptive accuracy based on a classic heartbeat counting task.

Individuals with poor interoceptive accuracy exhibited slower heartbeat following an acceleration in auditory stimulus rhythm. No such change was observed in individuals with good interoceptive accuracy. In the audio-visual condition, heart rate decreased when the speed of the stimuli was faster than baseline in both good and poor heartbeat perceivers. No significant results were observed in the visual condition.

If I understand well, the authors interpret these results in the framework of novelty detection (as induced by the change in sensory rhythm) and the heartbeat deceleration as an index of ‘acceptance’ and the heartbeat acceleration as expression of defensive reaction. The ‘acceptance’ is supposedly linked to the inability of making the distinction between heartrate and external stimulus rhythm of individuals with poor interoceptive accuracy.

The investigation of the cardiac activity in reaction to the timing of sensory stimuli is worthwhile, as we have limited knowledge of the mechanism reciprocally linking autonomic responses to sensory processing. The proposed experiments seem suitable to explore whether the change in the rhythm of external stimuli occurs in association to a change in cardiac activity. On the other hand, I am less convinced that the current design allows to relate interoceptive accuracy to the change in the autonomic response following auditory rhythm modification. This proposed association seems rather artificial and difficult to explain. The weakest argument is the assumption that poor interoception is associated to the difficulty in making a distinction between heartrate and stimulus rhythm. Interoception is measured based on a well-established counting task and I do not see how this accuracy is directly linked to the discrimination ability between own heartbeat and sensory stimuli rhythms. Likewise the notion of acceptance or defensive as expressed by the autonomic system behavior does not seem to have a strong scientific ground. In addition the manuscript is not clearly written.

The paper could be improved in several aspects for reaching the standard of a scientific publication.

6. PLOS authors have the option to publish the peer review history of their article (what does this mean?). If published, this will include your full peer review and any attached files.

Reviewer #1: No

Reviewer #2: No

---

## [Author Response · Author response to Decision Letter 0]

24 Apr 2021

Response: Thank you for your comment. We have checked the format again to ensure that our manuscript follows the style requirements before we submit it.

2. We note that the grant information you provided in the 'Funding Information' and 'Financial Disclosure' sections do not match.

When you resubmit, please ensure that you provide the correct grant numbers for the awards you received for your study in the 'Funding Information' section.

Response: We apologize for this mistake in the funding information. We have revised the text in the 'Funding Information' section.

Response: We apologize for the inconvenience that this may have caused. We have uploaded the relevant data as a supporting information. 

Reviewer #1: The study investigates if cardiac reactivity changes in response to manipulations in the external rhythm (auditory, visual, and audio-visual changes). The authors divide two groups in poor and good perceivers according to the heartbeat counting task results.

The idea quite interesting, the main aim was to study such changes in response to non-affective external stimuli, however the study presents a few weaknesses both theoretically and methodologically:

Response: We thank the reviewer for their careful reading and supportive comments. Your insightful comments and constructive suggestions have helped us improve our paper considerably. We have addressed each point below.

 

1) The introduction and the hypotheses could do a better job. The style could be a little more colloquial, it seems more like bullets points.

It is unclear: i) the rational in having faster and slower conditions from baseline and why only the faster condition seems to be affected. 

Response: We thank you for pointing out this important part of the study. In a previous study, the relationship between the heartbeat and the external stimulus was investigated, where the exteroceptive stimulus was slowed down [20, 22]. This particular study did not discover how autonomic reactivity affects the stimulus when the speed of the stimulus gradually increases. A previous study suggested that listening to music with a fast tempo increases the participants’ heart rate [21]. Therefore, we were interested in whether the heart rate changed when the stimulus’ speed gradually increased. As you pointed out in (ii), we might have expanded the scope of the study.

We hypothesized that heart rate could be linked to stimulus perception. People often perceive their heartbeat to have increased but rarely perceive it to have slowed down. Therefore, interoceptive accuracy may be influenced only when the stimulus becomes faster.

We have added text and references to make points more transparent in the introduction.

[Added text]

Page 3, line 72-84

Previous studies using periodic rhythm instead of music have only used stimuli with a decreasing tempo [20, 22]; we extended the tempo range to study the effect of stimuli with increasing tempos. A previous study has claimed that a decrease in the heart rate is observed when participants are exposed to a stimulus with a decreasing tempo [29]; therefore, a different effect may be seen when the tempo increases. The effect may be akin to listening to a tempo during exercise where the psychological effects are different than those in a resting state [30]. 

An increased tempo might have a different effect in decreasing cardiac tempo. Studies have not been consistent in elucidating how the heart rate changes when stimuli become faster [31, 32]. Participants often perceive the heartbeat to be faster but rarely perceive it to be slower. Therefore, interoceptive accuracy may be influenced only by a faster stimulus. For individuals with poor interoceptive accuracy, an accelerating stimulus would be expected to increase their heart rate. Because individuals with good interoceptive accuracy will notice this increase, their heart rate will not change.

 

ii) why three different modalities (one unimodal A and V and another bimodal AV) were tested and how a specific modality should be affected more or less than others. I wonder if using a single modality to increase the number of trials would have been a better way, rather than introducing two more modalities without very strong hypotheses and having few trials.

Response: Thank you for your comment. When we came up with the hypothesis, we thought that the heartbeat effect might differ upon the compounding the stimuli. For instance, reaction times to the stimulus are accelerated when stimuli from multiple modalities are presented simultaneously, which is termed as intersensory facilitation [34]. Many other studies have shown that multisensory integration affects mental processing [35]. However, these may not be related to heartbeats. In this case, we can observe if either the visual or auditory senses respond preferentially. For example, in the sound-induced flash illusion, visual information is influenced by auditory information [36]. If we can understand how these changes occur, we can determine which modality becomes representative for stimuli that mimic the heartbeat. Therefore, presenting the two modalities simultaneously is an essential part of this study. Nevertheless, the number of trials is a significant issue, although increasing the number of trials would be a burden for the participants.

We have added a sentence in the text about the significance of presenting the stimuli simultaneously. In addition, we mentioned the number of trials in the limitations section.

[Added text]

Page 4 line 88-95

In addition, the simultaneous presentation of visual and auditory stimuli may produce changes that are different from those of a single rhythm. For instance, reaction times to the stimulus were accelerated when multiple modalities are presented simultaneously, this is termed intersensory facilitation [34]. Many other studies have shown that multisensory integration affects mental processing [35]; however, it was not well discussed how multisensory integration effect to the tempo of the heartbeat. We therefore aimed to observe if either the visual or auditory senses respond preferentially. For example, in the sound-induced flash illusion, visual information is influenced by auditory information [36].

[Added text]

Page 14 line 329-331

Additionally, there were only two trials in this study, which were relatively small. Increasing the number of trials may produce more consistent results to show that interoceptive accuracy is related to autonomic reactivity.

 

iii) it seems that there was the aim to study differences in a group of people with high and low interoceptive accuracy that was not properly investigated at the step on statistical analysis (see point2).

I suggest to clarify all these points in the introduction/aims/hypothesis.

Response: Thank you for raising these concerns about the statistics. The main purpose of this study was to determine whether there was a difference between people with high and low interoceptive accuracy. As you pointed out, our statistical analysis may be less appropriate than the statistical analysis you suggested in your second comment. Therefore, we have re-analyzed our data according to your comments. The changes we made are described in the next paragraph (point 2).

 

2) Statistical Analysis: It is unclear how the 3 modalities (A/V/AV) were analyzed in the first ANOVA. If the aim was to investigate the differences between participants with higher and lower interoceptive accuracy and the modalities, I think that a single ANOVA design with group (poor and good perceivers) as between factor; speed (baseline, faster, slower) and stimulus type (A, V, AV) as within levels; would have been a better design. I wonder whether there was any other difference in the results between the two groups that was not reported. It seems that the two groups differ in the slower conditions, at the least in two modalities (A and AV).

I would suggest to perform a between subjects ANOVA 2Group x 3Speed x 3Stymulus Type, as I explained above. Once this has been clarified it will help the decision on the validity of the results.

Response: Thank you for your valuable suggestion regarding statistics. Based on your suggestion, we performed a 3-way ANOVA, and we obtained exciting results in addition to the previous results that interoceptive accuracy is related to changes in the heart rate. Therefore, we have rewritten the results and added a discussion of the new results.

[Added text]

Abstract line 13-20

Individuals with good interoceptive accuracy heart rate when the rhythm of the auditory stimulus changed, unlike individuals with poor interoceptive accuracy. Individuals with poor interoceptive accuracy could not compare their internal state to the external rhythm; therefore, it is thought that they recognize the stimulus as their heart rate. Individuals with good interoceptive accuracy are thought to distinguish their heart rate from the external rhythm. The modality difference was not observed in this study, which suggests that both visual and auditory stimuli help mimic heart rate.

[Added text]

Page 10, line 233-246

The average values of the IBI during the experimental tasks for each of the three conditions are shown in Table 2. We conducted a 3-way analysis of variance (ANOVA) to determine whether the difference in IBI was due to the difference in interoceptive accuracy (good or poor interoceptive accuracy), modality of the stimulus (auditory, visual, or compounding of auditory and visual), or the speed of the stimuli (baseline, faster than baseline, or slower than baseline). The degrees of freedom were adjusted using the Greenhouse–Geisser procedure. 

We found an interaction between interoceptive accuracy and the speed of the stimulus (F [1.92, 69.10] = 3.32, p < .05, η2 = .003). In the poor interoceptive accuracy group, heart rate increased when the stimulus speed was the increased cardiac tempo than when the stimulus speed was the decreased cardiac tempo, similar to the baseline heart rate (Fig 3a). In the good interoceptive accuracy group, the heart rate decreased when the stimulus speed get both faster or slower from the baseline, either faster or slower (Fig 3b). These results indicate that interoceptive accuracy is related to the autonomic response induced by non-affective rhythms.

[Added text]

Page 10, line 251-255

Figure 3 Interbeat intervals (IBIs) while presenting external rhythmic stimuli. In poor heartbeat perceivers, IBI increased when the speed of the stimuli was faster than their baseline heart rate. In good heartbeat perceivers, IBI increased when the speed of the stimuli was changed from their baseline heart rate. There were no significant differences among the modality of the stimulus.

[Added text]

Page 10, line 260-264

Individuals with poor interoceptive accuracy exhibited a change in heart rate when the rhythm of the stimulus was at the increased cardiac tempo. This tendency was not observed in individuals with good interoceptive accuracy. Instead, their heart rate became slower when the speed of the stimulus changed, regardless of whether it got faster or slower.

[Added text]

Page 12, line 289-297

For participants with good interoception, stimuli may be distinguished as being different from the heartbeat, yet remain a heartbeat-like stimuli; this is consistent with the findings of previous studies that have been predicted to induce relaxation [27]. Although it is difficult to evaluate the psychological effects of the stimuli in this study, we suggest that the parasympathetic nervous system became dominant during the viewing process. Conversely, this was only observed for individuals in the poor interoceptive accuracy group when the stimulus became faster. It may be easier to perceive one’s heartbeat when it is noticeably faster, for example, after exercise [46]. Therefore, it is likely that the stimulus's rhythm would be predicted as being the same as the heartbeat.

[Added text]

Page 13. Line 300-306

The results of the present study showed no difference in heart rate according to modality, contradicting the hypothesis. This suggests that both visual and auditory stimuli are useful for mimicking the heart rate, and that cross-modal effects [47, 48] do not occur—even when they are presented simultaneously. Although the perceived ease of change was different depending on the type of stimulus, it was not affected by the type of stimulus, further suggesting that the ability to compare the rhythm and interoception is more important than the ease of conscious perception of the stimulus. 

3) The method used is my biggest concern, it seems that the conditions (faster and slower) were manipulated according to the cardiac rhythm registered at baseline offline (prior the task execution). Therefore the external stimuli were not perfectly synchronized (slower or faster in 30 ms) with the real rhythm online at the time of the task execution. I fatigue to accept the validity of their results, as the real cardiac rhythm could have been changed during the task execution and not recorded, in this way the external stimuli could never adjust accordingly.

Add this as a limitation of the study.

Response: Thank you for raising these concerns. As you mentioned, rhythm was determined as the participants’ heart rates measured at an offline baseline period. We chose to use this method because we were interested in how autonomic reactivity was affected when the rhythm of the external stimuli changed. If the r-wave triggered the stimulus, the stimuli would be synchronous to the heart rate, but this would reduce the consistency of the rhythm. Because cardiac rhythm has variability, if the rhythm of the stimulus changes by heartbeat, it is difficult to discriminate whether the changes in the heart rate is caused by the variability or by the experimental manipulation. However, it is true that the rhythm of the heartbeat is different from that of the actual heartbeat, and your criticism cannot be denied. The phenomenon that might be observed when the r-wave triggered the rhythm is fundamental, although it might differ from what we expected to observe. However, there could be better ways to determine the stimulus's speed than using the offline baseline heart rate, such as using the participant's cardiac r–r interval at the beginning of the trial.

We have added the limitation of this procedure as follows.

[Added text]

Page 14, line 320-329

Some limitations of our current work must be considered to help design future studies. The baseline in this study referred to the mean resting heart rate, which was not recorded at the beginning of the task. Therefore, the r–r interval could have been different from the baseline when the participants started their trials. Even if stimuli were presented synchronously with the heartbeat, it is unclear whether the participants would perceive this as the heartbeat, and fluctuations in a heartbeat could erase experimental manipulation changes. Because synchronizing with the heartbeat creates various constraints, this study aimed to make steady changes in the stimuli speed. Future studies may benefit from making the baseline rhythm closer to the heartbeat, for example, to better match the tempo of the stimuli to that of the heart rate at the start of the trial.

 

Minor points:

1) Page 1 line 22: specify r-r

Response: We apologize for this inconvenience. We specified the cardiac r–r interval and added a description of it in the relevant section.

[Added text]

Page 1 Line 22-24

Autonomic reactivity was assessed when participants were exposed to an external stimulus, based on changes in the cardiac r–r interval and the distance between the two r waves on the electrocardiogram [17, 18].

2) Page 5 line 110: I wonder if this is a typo and line 110 should be: (1/6 Σ(1 − (|actual heartbeats − reported heartbeats|/actual heartbeats)).

Response: We apologize for the confusion. The equation in our manuscript was for a single trial, and we had written in the text that the average of six trials was used as the index. Because the expression you suggested is easier to understand, we changed the expression as follows:

[Added text]

Page 6 line 144 - 146

We calculated the score of the heartbeat counting task using the following formula: (1/6 Σ (1 − (|actual heartbeats − reported heartbeats|/actual heartbeats)).

3) It is unclear the total number of trials: 6 trials for each condition (baseline - shorter and longer interval?) or 6 trials for each modality (A V AV) ? Please specify.

Response: We apologize for the inadequate description. There were six types of trials in this experiment: visual stimulus got faster, visual stimulus got slower, auditory stimulus got faster, auditory stimulus got slower, the visual received auditory stimuli were presented and they got faster, and the visual and auditory stimuli were presented and they became slower. Each trial was conducted twice. We have revised the manuscript to clarify the number of trials.

[Added text]

Page 7 line 168-173

Participants conducted six types of conditions: increasing frequency of visual stimuli; decreasing frequency of visual stimuli; increasing frequency of auditory stimuli; decreasing frequency of visual stimuli; increasing frequency of visual and auditory stimuli; and decreasing frequency of visual and auditory stimuli. These conditions were combined into one group, and each of the six conditions was performed twice, as shown in Table 2.

4) I would avoid any discussion relating brain activity as the study is purely behavioral. Please remove.

Response: Thank you for your suggestion. As you pointed out, our study is purely behavioral, and it should be more prudent to have a discussion about the brain. We have removed the comment about the brain in the discussion.

[Added text]

Page 15 line 352-353

Further studies are necessary to assess the explicit expression of these interactions in our mental state.

 

Reviewer #2: The authors investigate the cardiac reactivity in healthy participant by recording ECG during passive presentation of auditory, visual stimuli or both. Sensory stimuli followed a rhythm closely matching participants' heartbeat and then gradually accelerating or decelerating. Participants were separately screened in order to evaluate their interoceptive accuracy based on a classic heartbeat counting task.

Individuals with poor interoceptive accuracy exhibited slower heartbeat following an acceleration in auditory stimulus rhythm. No such change was observed in individuals with good interoceptive accuracy. In the audio-visual condition, heart rate decreased when the speed of the stimuli was faster than baseline in both good and poor heartbeat perceivers. No significant results were observed in the visual condition.

If I understand well, the authors interpret these results in the framework of novelty detection (as induced by the change in sensory rhythm) and the heartbeat deceleration as an index of 'acceptance' and the heartbeat acceleration as expression of defensive reaction. The 'acceptance' is supposedly linked to the inability of making the distinction between heartrate and external stimulus rhythm of individuals with poor interoceptive accuracy.

Response: We thank you for your thoughtful comments and for raising an important concern, as you have helped us significantly improve our manuscript. We have addressed your points below.

On the other hand, I am less convinced that the current design allows to relate interoceptive accuracy to the change in the autonomic response following auditory rhythm modification. This proposed association seems rather artificial and difficult to explain. The weakest argument is the assumption that poor interoception is associated to the difficulty in making a distinction between heartrate and stimulus rhythm. Interoception is measured based on a well-established counting task and I do not see how this accuracy is directly linked to the discrimination ability between own heartbeat and sensory stimuli rhythms. Likewise the notion of acceptance or defensive as expressed by the autonomic system behavior does not seem to have a strong scientific ground. 

Response: 

We thank you for raising these critical concerns and providing us these comments. We have a confidence that previous research provides sufficient evidence to hypothesize that interoceptive accuracy is related to the discrimination ability between heartbeat and sensory stimuli rhythms. The heart rate discrimination task, which is a measure of interoceptive accuracy, compares the rhythm of external stimuli with the actual heart rate [23]. A person with a poor score will perceive a different rhythm as their heartbeat, which means that the exteroceptive sensation induces their interoception [24]. We were aware that there is an argument that the heartbeat discrimination task and the heartbeat counting task measures are unrelated [54]. However, because these two tasks have a commonality that participants have to recognize the rhythm of their heartbeat, it can be said that the heartbeat counting task is related to discrimination ability [9]. Moreover, because they have to recognize that the rhythm became increase or decrease, it might be possible that the heartbeat detection task is more related to this discrimination ability if the heartbeat is continuously monitored. We had enough base of an argument to hypothesize that the interoception accuracy and the ability to discriminate between heartbeats and external stimuli are related.

However, the lack of a solid and stable physiological background, including the previous studies on which our research is based, cannot be denied. We should regard the background and foundation as relatively weak. In particular, it is premature to link the change in heart rate to the acceptance of stimuli or defensive reactions. We regard our study only to mention the parasympathetic activity related to the relaxation that was pointed out in previous research [22, 27]. Heart rate deceleration is the same phenomenon as the orient response [46]. Therefore, because stimuli were similar, connecting these ideas would help to accumulate the physiological background. We conducted this study in the hope that it would help strengthen the weak biological background. As a result, we obtained evidence to support our hypothesis, and we believe that further research will help in investigating the underlying biological mechanisms.

[Added text]

Page 2, line 46-58

External stimuli in which the rhythm approximates the heartbeat are often treated as a representation of the heartbeat in medical setting or researches about the interoception; a typical example is the heartbeat discrimination task. Similar to the heartbeat counting task, this task is treated as a measure of interoceptive accuracy [23]. Although it is has been pointed out that this task is difficult [26], it enables the observation of individual differences, and both heartbeat counting task and heartbeat discrimination task are proper indicators of interoceptive accuracy [24]. Therefore, the ability to distinguish an external stimulus's tempo from that of the heartbeat is essential for accurately capturing the interoception. A previous study has shown that the perception of cardiac interoception is sensitive to external rhythms [25]. The ability to discriminate between one’s heartbeats and external stimuli by interoceptive sensations depends on ignoring other exteroceptive information, thereby allowing appropriate mental state adjustments. In this case, the autonomic response may vary depending on the interoceptive accuracy.

[Added text]

Page 12, line 289-297

For participants with good interoception, stimuli may be distinguished as being different from the heartbeat, yet remain a heartbeat-like stimuli; this is consistent with the findings of previous studies that have been predicted to induce relaxation [22, 27]. Although it is difficult to evaluate the psychological effects of the stimuli in this study, we suggest that the parasympathetic nervous system became dominant during the viewing process. Conversely, this was only observed for individuals in the poor interoceptive accuracy group when the stimulus became faster. It may be easier to perceive one’s heartbeat when it is noticeably faster, for example, after exercise [46]. Therefore, it is likely that the stimulus's rhythm would be predicted as being the same as the heartbeat.

[Added text]

Page 15 344-348

While there is a robust relationship between emotional stimuli and internal receptive sensations, a solid scientific background is lacking since no emotional information is involved. The results of our study suggest that interoception commits to the perception of processing stimuli, as well as other exteroceptive sensations.

In addition the manuscript is not clearly written.

Response: We apologize for the lack of clarity in the manuscript. Based on your comment, we have clarified the theoretical part and its limitations, and for the English aspect, we have clarified the expressions by taking a proofreading of native speaker.

---

## [Decision Letter · Decision Letter 1]

27 May 2021

PONE-D-20-20821R1

Effects of Interoceptive Accuracy in Autonomic Responses to External Stimuli Based on Cardiac Rhythm

PLOS ONE

Dear Dr. Tanaka,

Thank you for submitting your manuscript to PLOS ONE. After careful consideration, we feel that it has merit but does not fully meet PLOS ONE’s publication criteria as it currently stands. Therefore, we invite you to submit a revised version of the manuscript that addresses the points raised during the review process.

Effect size is still a major concern.  Additional works such as statistical cross-validation analyses based on the suggestions from the reviewer may be needed to improve the manuscript further. 

We look forward to receiving your revised manuscript.

Kind regards,

Zhishun Wang, Ph.D.

Academic Editor

PLOS ONE

Journal Requirements:

Reviewers' comments:

Reviewer's Responses to Questions

**Comments to the Author**

1. If the authors have adequately addressed your comments raised in a previous round of review and you feel that this manuscript is now acceptable for publication, you may indicate that here to bypass the “Comments to the Author” section, enter your conflict of interest statement in the “Confidential to Editor” section, and submit your "Accept" recommendation.

Reviewer #1: (No Response)

Reviewer #2: All comments have been addressed

2. Is the manuscript technically sound, and do the data support the conclusions?

Reviewer #1: No

Reviewer #2: Yes

3. Has the statistical analysis been performed appropriately and rigorously? 

Reviewer #1: Yes

Reviewer #2: Yes

4. Have the authors made all data underlying the findings in their manuscript fully available?

Reviewer #1: Yes

Reviewer #2: Yes

5. Is the manuscript presented in an intelligible fashion and written in standard English?

Reviewer #1: Yes

Reviewer #2: Yes

6. Review Comments to the Author

Reviewer #1: I would like to thank the authors for responding exhaustively to my comments. They also reorganized the manuscript in a way that reads better now.

However, although I appreciate the work that they put into the new version of the manuscript and the experimental question that is novel, I still have some methodological issues that cannot be solved at this stage:

1) The numbers of trials: There are only 2 trials per condition, and this makes the study very weak.

2) The results from the 3-way ANOVA -that I asked to perform in the first revision- shows one significant effect that the authors report as p<0.05 and η2 = .003 that is a very small effect size; I wonder if there is an error here. Since the effect size is very small for a significant effect.

3) The baseline that was not recorded at the beginning of the task.

My conclusion is that I cannot accept the manuscript in this form. One possibility would be to collect more data, this may compensate for the small numbers of trials. I would recommend them to perform an a-priori power analysis based on literature on similar effects. I think that increasing the number of participants would be informative on the effects that the authors are looking for that seem to have a small effect size for now.

Reviewer #2: The authors improved to a great extent the overall manuscript quality and I am happy to endorse its publication.

7. PLOS authors have the option to publish the peer review history of their article (what does this mean?). If published, this will include your full peer review and any attached files.

Reviewer #1: No

Reviewer #2: No

---

## [Author Response · Author response to Decision Letter 1]

10 Jul 2021

We would like to thank the editor and reviewers for their thoughtful comments and advice, which have helped us make appropriate revisions to our manuscript. We also would like to express our deep gratitude to Reviewer 2 for endorsing the publication of our manuscript. Here, we have provided point-by-point responses to these comments and have revised our manuscript accordingly.

To the Associate editor:

Effect size is still a major concern. Additional works such as statistical cross-validation analyses based on the suggestions from the reviewer may be needed to improve the manuscript further.

Response: We have taken your concern that the effect size is small seriously and we respect your opinion that we have not secured a sufficient number of participants. Adding more participants was an option to be considered, but laboratory conditions are different now than they were when we conducted this experiment, and it is difficult to add experimental data in the exact same setting. In addition, the COVID-19 pandemic made it extremely difficult to conduct the experiment. We concluded that additional data are not a realistic option for us. On the other hand, the power analysis and bootstrap tests suggested that although it was not high, the value of pη2 was above the minimum size. Therefore, we would like to respond to this comment by adding the following information to the manuscript. We would appreciate your understanding. 

[Added text]

Page 5, line 114

A priori power analysis indicated that a sample size of at least 28 was necessary.

Page 10, line 242

(F [1.92, 69.10] = 3.32, p < .05, pη2 = .10, CI - .01 - .20).

Page 14, line 321-322

This study was only able to show minimal effect size. Increasing the research size and increasing the sample size would be necessary to establish this effect as robust.

Response: Thank you for the opportunity to check the references. We reviewed the reference list again and confirmed that no papers had been retracted.

Reviewer #1: 

I would like to thank the authors for responding exhaustively to my comments. They also reorganized the manuscript in a way that reads better now.

 However, although I appreciate the work that they put into the new version of the manuscript and the experimental question that is novel, I still have some methodological issues that cannot be solved at this stage:

Response: We are grateful for the evaluation of the revised manuscript. We would also like to thank you for your constructive comments and suggestions. We have addressed each point below.

1) The numbers of trials: There are only 2 trials per condition, and this makes the study very weak.

Response: Thank you for your comment. This experiment had only two trials for each condition. This is because previous studies included only one trial in the experiments (Iwanaga et al., 2005; Takenaka et al., 2005), and we thought it was better to take as much time as possible for each trial rather than increasing the number of trials. We believe that we could show some effects, even with just two trials. In addition, there was a problem of participants getting fatigued during the experiment. In the present experiment, we alternated the experimental task with a heartbeat detection task or a time estimation task. This method was used to reduce the effects of habituation on continuous stimuli and provide a reliable time for the participants’ heart rates to return to a steady state after the stimuli changed. Therefore, the duration of experiment was longer, with each trial lasting 20-25 min. We believe that increasing the number of trials would further increase the burden on the participants.

[Added text] 

Page 14-15, line 336-342

Moreover, there were only two trials in this study, and these were relatively small. Since only a single trial was conducted for each condition in previous studies [21, 43], it was considered more important to take as much time as possible for one trial than to increase the number of trials. In addition, a long experiment may result in increased fatigue of the participants, which may affect their heart rates. Therefore, in the future, increasing the number of trials by dividing the experiment into multiple days may produce results that are more consistent and demonstrate that interoceptive accuracy is related to autonomic reactivity.

2) The results from the 3-way ANOVA -that I asked to perform in the first revision- show one significant effect that the authors report as p<0.05 and η2 = .003 which is a very small effect size; I wonder if there is an error here. Since the effect size is very small for a significant effect.

Response: Thank you for raising this concern. We rechecked our statistical calculations and determined that η2 was most likely inappropriate for this experiment. Increasing the sample size is the best way to increase the effect size, but the situation in our laboratory and across the world did not allow for this. We re-calculated pη2 and found it to be .10. This further validates the trustworthiness of our experimental results. In addition, the confidence interval for the effect size using the bootstrap method was .01–.20, which we believe increases the reliability of the results.

[Added Text] 

Page 10, line 242

(F [1.92, 69.10] = 3.32, p < .05, pη2 = .10, CI - .01 - .20).

3) The baseline that was not recorded at the beginning of the task.

Response: Thank you for your essential remarks. As you stated, we did not calculate the speed of the stimulus based on the baseline before the trial. If we assume that the baseline is before the start of the trial, we can calculate the heart rate easily since we measured the participants’ heart rates before the trial. However, there is a possibility that there was an effect from the previous task on participants’ heart rates, and if we calculate the average, although it is supposed to be closer to the actual heart rate than the resting baseline, it is still not synchronized with the actual heart rate. Therefore, we believe that it is best to compare heart rates during stimulus presentation. In addition, our results are based on responses to heartbeat-like stimuli and not on synchronization with actual heartbeats. Therefore, we believe that our results are meaningful even if we did not measure the baseline. While it is tempting to align the heart rates with the baseline, this would be distracting from our goal of examining the relationship between rhythmic sound presentation and heart rate. We hope that you will allow us to finalize this experimental condition.

[Added text] 

Page 14-15, line 331 - 336

In addition, the study did not compare the baseline heart rate and the condition in which the stimulus was presented. Although there was a significant difference between these states, the possibility that interoceptive accuracy is involved in switching the state with modality and the state without modality cannot be denied. In future studies, it will be important to increase the inter-trial interval and compare the results with the baseline before the stimulus presentation and between trials.

My conclusion is that I cannot accept the manuscript in this form. One possibility would be to collect more data, this may compensate for the small numbers of trials. I would recommend them to perform an a-priori power analysis based on literature on similar effects. I think that increasing the number of participants would be informative on the effects that the authors are looking for that seem to have a small effect size for now.

Response: Thank you for this suggestion. We performed a power analysis, as suggested by the reviewer. We used the traditional effect size of 0.25 because there were no similar results showing effect sizes in previous studies. The results indicated that a reasonable sample size would be 28, which is more than the current number of participants. Due to the miscalculation of the effect size, we are hopeful that this is a reasonable sample size for our results. Increasing the sample size is also very attractive, and we understand that it is an efficient way to increase the effect size. We also considered adding more participants. However, laboratory conditions are different now than they were when we conducted this experiment, and it is difficult to add experimental data in the exact same setting. In addition, the COVID-19 pandemic made it extremely difficult to conduct the experiment. Therefore, we concluded that adding more data is not a realistic option. Since we had an effect size of value of pη2 that was above the minimum threshold, we would like to respond to this comment by adding the following to the manuscript. We would appreciate your understanding.

[Added text]

Page 5, line 114

A priori power analysis indicated that a sample size of at least 28 was necessary.

Page 14, line 321-322

This study was only able to show a minimal effect size. Increasing the research size and increasing the sample size would be necessary to establish this effect as robust.

---

## [Decision Letter · Decision Letter 2]

22 Jul 2021

PONE-D-20-20821R2

Effects of Interoceptive Accuracy in Autonomic Responses to External Stimuli Based on Cardiac Rhythm

PLOS ONE

Dear Dr. Tanaka,

Thank you for submitting your manuscript to PLOS ONE. After careful consideration, we feel that it has merit but does not fully meet PLOS ONE’s publication criteria as it currently stands. Therefore, we invite you to submit a revised version of the manuscript that addresses the points raised during the review process.

We look forward to receiving your revised manuscript.

Kind regards,

Zhishun Wang, Ph.D.

Academic Editor

PLOS ONE

Journal Requirements:

Reviewers' comments:

Reviewer's Responses to Questions

**Comments to the Author**

1. If the authors have adequately addressed your comments raised in a previous round of review and you feel that this manuscript is now acceptable for publication, you may indicate that here to bypass the “Comments to the Author” section, enter your conflict of interest statement in the “Confidential to Editor” section, and submit your "Accept" recommendation.

Reviewer #1: (No Response)

2. Is the manuscript technically sound, and do the data support the conclusions?

Reviewer #1: Partly

3. Has the statistical analysis been performed appropriately and rigorously? 

Reviewer #1: No

4. Have the authors made all data underlying the findings in their manuscript fully available?

Reviewer #1: Yes

5. Is the manuscript presented in an intelligible fashion and written in standard English?

Reviewer #1: Yes

6. Review Comments to the Author

Reviewer #1: All comments from authors have been addressed. However, I want to ask the authors to add the details of their power analysis.

They write:

Page 5, line 114

A priori power analysis indicated that a sample size of at least 28 was necessary

Only this sentence is not enough to provide evidence of a well-performed power analysis. The authors need to report in the paper exactly how they performed this analysis.

For instance: type of test used, number of the group, correlation, alpha level, power and the effects size used.

If the effects size used to calculate the power is based on literature, the authors need to indicate the effects size used according to literature and cite those papers.

What kind of design did they indicate in this analysis? How much power,correlation and effect size they indicated to obtain that a sample size of 28 was big enough to support their effects?

All these details are missing and I fatigue again to accept the manuscript in this shape.

7. PLOS authors have the option to publish the peer review history of their article (what does this mean?). If published, this will include your full peer review and any attached files.

Reviewer #1: No

---

## [Author Response · Author response to Decision Letter 2]

10 Aug 2021

We would like to express our deepest gratitude to the editor and reviewer. The suggestions we have received have helped us to improve our manuscript.

To the Associate editor:

Response: Thank you for the opportunity to check the figures. We used the suggested website and modified to meet PLOS requirements.

Reviewer #1: 

All comments from authors have been addressed. However, I want to ask the authors to add the details of their power analysis.

They write:

Page 5, line 114

A priori power analysis indicated that a sample size of at least 28 was necessary

Only this sentence is not enough to provide evidence of a well-performed power analysis. The authors need to report in the paper exactly how they performed this analysis.

For instance: type of test used, number of the group, correlation, alpha level, power and the effects size used.

If the effects size used to calculate the power is based on literature, the authors need to indicate the effects size used according to literature and cite those papers.

What kind of design did they indicate in this analysis? How much power, correlation and effect size they indicated to obtain that a sample size of 28 was big enough to support their effects?

All these details are missing and I fatigue again to accept the manuscript in this shape.

Response: We apologize that our description was inadequate. We added in our manuscript that we conducted our power analysis by the following study design, F-test with two groups with three measurements, and parameters as alpha = .05, power = 0.80, and correlation = 0.50. Since previous studies did not describe the effect sizes in their papers, the traditional effect size of 0.25 was used in our research. We cited Cohen's criteria to determine the effect size. We also added information about the software to calculate the result.

[Added text]

Page 5, line 114-118

We performed a priori power analysis for the sample size estimation for F-test based on our study design, two groups with three measurements, using GPower 3.1 [38]. Assuming that this study has a medium effect size, we used 0.25 based on Cohen's criteria [39]. With an alpha = .05, power = 0.80, and correlation = 0.50, it indicated that a sample size of at least 28 was necessary.

---

## [Decision Letter · Decision Letter 3]

19 Aug 2021

Effects of Interoceptive Accuracy in Autonomic Responses to External Stimuli Based on Cardiac Rhythm

PONE-D-20-20821R3

Dear Dr. Tanaka,

We’re pleased to inform you that your manuscript has been judged scientifically suitable for publication and will be formally accepted for publication once it meets all outstanding technical requirements.

Kind regards,

Zhishun Wang, Ph.D.

Academic Editor

PLOS ONE

Additional Editor Comments (optional):

Reviewers' comments:

Reviewer's Responses to Questions

**Comments to the Author**

1. If the authors have adequately addressed your comments raised in a previous round of review and you feel that this manuscript is now acceptable for publication, you may indicate that here to bypass the “Comments to the Author” section, enter your conflict of interest statement in the “Confidential to Editor” section, and submit your "Accept" recommendation.

Reviewer #1: All comments have been addressed

2. Is the manuscript technically sound, and do the data support the conclusions?

Reviewer #1: Yes

3. Has the statistical analysis been performed appropriately and rigorously? 

Reviewer #1: Yes

4. Have the authors made all data underlying the findings in their manuscript fully available?

Reviewer #1: Yes

5. Is the manuscript presented in an intelligible fashion and written in standard English?

Reviewer #1: Yes

6. Review Comments to the Author

Reviewer #1: The authors responded properly to all my comments. I do not have further comments and I accept the paper in its actual shape.

7. PLOS authors have the option to publish the peer review history of their article (what does this mean?). If published, this will include your full peer review and any attached files.

Reviewer #1: No

---

## [Editor Report · Acceptance letter]

23 Aug 2021

PONE-D-20-20821R3 

Effects of interoceptive accuracy in autonomic responses to external stimuli based on cardiac rhythm 

Dear Dr. Tanaka:

I'm pleased to inform you that your manuscript has been deemed suitable for publication in PLOS ONE. Congratulations! Your manuscript is now with our production department. 

Kind regards, 

on behalf of

Dr. Zhishun Wang 

Academic Editor

PLOS ONE